# Nutritional Status Is Not a Predictor of Anaphylaxis Severity in a Pediatric Cohort: A Retrospective Analysis

**DOI:** 10.3390/nu17183023

**Published:** 2025-09-22

**Authors:** Izabela Kucharek, Krzysztof Przystał-Dyszyński, Aleksandra Godyńska, Maria Gregorczyk, Adam J. Sybilski

**Affiliations:** Clinical Department of Pediatrics and Allergology, National Medical Institute of the Ministry of the Interior and Administration, 02-507 Warsaw, Polandadam.sybilski@pimmswia.gov.pl (A.J.S.)

**Keywords:** anaphylaxis, body mass index, child, pediatric obesity, nutritional status, severity of illness index, retrospective studies

## Abstract

**Background**: Childhood obesity is a pro-inflammatory state associated with poorer outcomes in chronic allergic diseases, such as asthma, and in adults, it is a recognized risk factor for more severe anaphylaxis. However, whether this association extends to the pediatric population remains unclear. **Objectives**: The aim of this study was to assess the association between nutritional status, as measured by Body Mass Index (BMI), and anaphylaxis severity and presentation in a cohort of hospitalized children. **Methods**: We retrospectively assessed the association between BMI categories (underweight, normal weight, overweight, and obese) and the severity (WAO grading) and clinical presentation of anaphylaxis in 199 hospitalized children (0–18 years). **Results**: No statistically significant association was found between BMI categories and anaphylaxis severity (χ^2^ = 7.06, *p* = 0.861). Severe reactions (WAO grades 4–5) were rare across BMI categories, occurring in 0% of underweight, 3.8% of normal-weight, 9.1% of overweight, and 7.7% of obese children. In regression analyses adjusting for age, sex, asthma, and atopic dermatitis, BMI was not an independent predictor of anaphylaxis severity, whether considered as a categorical or continuous variable (all odds ratios non-significant, 95% CIs crossing 1). Similarly, organ system involvement did not differ between BMI groups (all *p* > 0.05). **Conclusions**: In this pediatric cohort, contrary to findings in adults, we did not find nutritional status to be a predictor of anaphylaxis severity or presentation. This suggests obesity’s role as a risk factor may be age-dependent and that adult data should be extrapolated to children with caution.

## 1. Introduction

Childhood obesity and asthma are among the most common chronic health conditions in pediatric populations, and both have shown marked increases in prevalence over recent decades [1]. Their co-occurrence has increasingly attracted scientific interest, partly due to overlapping immunological and metabolic mechanisms that may underlie both conditions [2,3].

A review of epidemiological data indicates that, although anaphylaxis is less prevalent in children than in adults, it remains a clinically significant condition. A recent population-based meta-analysis estimated the overall incidence of anaphylaxis at approximately 46 cases per 100,000 person-years, with higher rates reported in Europe (71/100,000). In the pediatric population, the incidence was considerably lower, with approximately five cases per 100,000 children per year (incidence rate ratio 0.1 compared with the general population) [4]. Earlier systematic reviews have reported a much broader range, from 1 to 761 cases per 100,000 person-years, reflecting substantial heterogeneity across study designs and geographical regions [5]. Despite the fact that the overall mortality rate remains low (i.e., less than 1%), severe courses requiring admission to an intensive care unit are observed in approximately 10% of pediatric cases [6]. The findings, when considered as a whole, emphasize the significant clinical impact of pediatric anaphylaxis and underscore the necessity for additional research to be conducted into the factors that may contribute to its severity. Simultaneously, the global prevalence of childhood overweight and obesity has continued to rise over the past decades, affecting a substantial proportion of the pediatric population worldwide [7]. In the search for potential modifiers of anaphylaxis severity, pediatric obesity merits particular attention, not only because of its growing prevalence worldwide but also due to its characterization as a systemic pro-inflammatory state, marked by increased levels of pro-inflammatory cytokines and adipokines and a concomitant decrease in regulatory mediators [2].

These immunological alterations have been suggested to influence allergic disease susceptibility and expression, particularly in asthma [2,3] and could potentially play a role in acute allergic reactions.

Obesity has also been shown to negatively affect respiratory mechanics by reducing lung volumes and thoracic compliance and increasing airway resistance [8,9]. Specifically, obesity-related dysanapsis—a mismatch between airway diameter and lung volume—has been identified as a potential mechanism behind increased bronchial reactivity and airway narrowing in affected children [8].

In the context of asthma, overweight and obesity have been demonstrated to be associated with a heightened frequency of symptoms, diminished disease control, and increased medication use, irrespective of age or sex [3]. However, while the majority of research focuses on chronic allergic conditions, the influence of body composition on the clinical presentation of anaphylaxis—an acute, life-threatening hypersensitivity reaction—remains to be elucidated. Although obesity has been identified as an independent risk factor for more severe and even fatal anaphylaxis in adult populations [10,11], the potential link between nutritional status and reaction severity in children remains insufficiently explored.

We therefore hypothesized that childhood overweight and obesity act as modifiers of anaphylaxis severity, shaping both the intensity and the organ-specific profile of reactions. To test this hypothesis, we assessed the relationship between nutritional status—measured via body mass index (BMI)—and the clinical presentation and severity of anaphylaxis in children hospitalized at the Department of Pediatrics and Allergology, Central Clinical Hospital of the Ministry of the Interior and Administration in Warsaw. Specifically, we analyzed whether the frequency of symptoms from specific organ systems, as well as the overall severity of anaphylaxis, differed across BMI categories defined using the Centers for Disease Control and Prevention (CDC) BMI-for-age growth charts.

## 2. Materials and Methods

### 2.1. Study Design

This was a retrospective, single-center observational study designed to assess the relationship between nutritional status and the clinical presentation of anaphylaxis in a pediatric population.

### 2.2. Setting

The study was based on a retrospective analysis of medical records from patients hospitalized at the Department of Pediatrics and Allergology of the National Medical Institute of the Ministry of the Interior and Administration in Warsaw. The analysis included patient data collected from January 2021 to June 2025.

### 2.3. Participants

The study protocol included all patients aged 0–18 years hospitalized with a diagnosis of anaphylaxis. For the final analysis, only patients who met all of the following inclusion criteria were enrolled:A clinical diagnosis of anaphylaxis consistent with the EAACI 2021 guidelines [12].Complete medical records of the anaphylactic episode, allowing for a clear assessment of the clinical presentation and classification of severity.Availability of complete anthropometric data (body weight and height) necessary for Body Mass Index (BMI) calculation.

Patients were excluded if any of the following applied:Hospitalization was for conditions other than anaphylaxis.Clinical data were incomplete or insufficient to apply the EAACI 2021 diagnostic criteria or to classify severity.Anthropometric measurements required for BMI calculation (weight and height) were unavailable.

### 2.4. Definitions and Variables

Anaphylaxis diagnosis: Anaphylaxis was defined according to the European Academy of Allergy and Clinical Immunology (EAACI) 2021 criteria [12], requiring acute onset of symptoms with skin/mucosal involvement and either respiratory compromise or cardiovascular instability, or the involvement of ≥2 organ systems after exposure to a likely allergen. Classification was performed retrospectively based on hospital medical records by two independent investigators, with discrepancies resolved by consensusAnaphylaxis Severity Grading: The severity of the reaction was assessed using the 5-grade system for systemic allergic reactions proposed by the World Allergy Organization (WAO) Anaphylaxis Committee [13].Exposure Variable: The primary exposure variable was the patient’s nutritional status. It was determined using the Body Mass Index (BMI), calculated as weight in kilograms divided by the square of height in meters (kg/m^2^). Patients were then categorized based on the age- and sex-specific BMI percentiles from the Centers for Disease Control and Prevention (CDC) growth charts into four groups: underweight (<5th percentile), normal weight (5th to <85th percentile), overweight (85th to <95th percentile), and obesity (≥95th percentile).Outcome Variables: The primary outcome variables were the clinical presentation and severity of anaphylaxis.Clinical Presentation: Assessed based on the recorded presence or absence of symptoms in five organ systems: cutaneous, respiratory, cardiovascular, gastrointestinal, and neurological.Anaphylaxis Severity: Graded from 1 to 5 according to the WAO grading system [13].Comorbidities: Asthma and atopic dermatitis were collected as baseline characteristics in order to describe the study population and assess potential confounders. Information on these conditions was obtained from the medical history provided by caregivers and verified, when available, by prior hospital records.

### 2.5. Study Size

The sample size was determined by the total number of eligible cases available in the hospital’s medical database during the study period. All patients hospitalized between January 2021 and June 2025 who met the pre-defined inclusion criteria were included in the analysis. No formal sample size calculation was performed beforehand. To address this limitation, we conducted post hoc power analyses for the main analysis (four BMI categories × WAO severity grades) as well as for additional sensitivity analyses with aggregated BMI categories (underweight/normal weight vs. overweight/obese), with and without collapsing severity into mild (grades 1–3) vs. severe (grades 4–5).

### 2.6. Statistical Methods

Descriptive statistics were used to summarize the characteristics of the study population. The analysis of the relationship between nutritional status (BMI categories) and the clinical presentation of anaphylaxis was performed in three stages:

First, the association between BMI categories and the overall severity of anaphylaxis was assessed using the chi-squared (χ^2^) test. Furthermore, post hoc power analyses were conducted for this primary analysis (four BMI categories x five WAO severity grades) and for two additional sensitivity analyses. The first approach involved the aggregation of BMI into two categories (underweight/normal weight vs. overweight/obese), whilst retaining the full WAO grading. The second approach involved the same BMI aggregation with severity collapsed into mild (grades 1–3) vs. severe (grades 4–5). Subgroup-specific post hoc powers were also calculated. Additionally, multivariate logistic regression analyses were conducted to evaluate whether BMI was independently associated with anaphylaxis severity after adjusting for age, sex, asthma, and atopic dermatitis.

Second, to assess whether the frequency of symptoms differed between the four BMI categories, the chi-squared (χ^2^) test was used for each organ system (cutaneous, respiratory, gastrointestinal, cardiovascular, and neurological). Post hoc pairwise comparisons between BMI groups were conducted where appropriate, with a Bonferroni-corrected significance level set at *p* < 0.0083.

Third, to analyze the frequency distribution of symptoms within each BMI category, Cochran’s Q test was performed. This test determined whether, for a given BMI category, symptoms from different organ systems occurred with significantly different frequencies. For significant results from Cochran’s Q test, post hoc pairwise comparisons between organ systems were performed using McNemar’s test, with a Bonferroni-corrected significance level set at *p* < 0.005. For all primary analyses, a *p*-value of less than 0.05 was considered statistically significant. For all the above analyses, calculations were performed using the Python 3.12.3 programming language with the following packages: statsmodels 0.14.5, scipy 1.16.0, pandas 2.3.1.

## 3. Results

### 3.1. Participant Characteristics

A total of 199 pediatric patients who met the inclusion criteria were included in the final analysis. The cohort was categorized based on nutritional status into four groups: underweight (n = 6, 3.0%), normal weight (n = 158, 79.4%), overweight (n = 22, 11.1%), and obese (n = 13, 6.5%). Male sex accounted for nearly 70% of the overall cohort, with the highest proportion observed in the obese subgroup (92.3%). Comorbid atopic diseases were common: 22.6% of patients had asthma, while 47.7% were diagnosed with atopic dermatitis. The demographic and clinical characteristics of the study cohort are presented in Table 1.

There were no statistically significant differences in the baseline distribution of age, sex, or the prevalence of comorbid atopic diseases (asthma and atopic dermatitis) between the four BMI groups. This indicates a homogenous baseline across the compared groups.

### 3.2. Association Between Nutritional Status and Anaphylaxis Severity

The primary analysis revealed no statistically significant association between BMI category and the severity grade of anaphylaxis (χ^2^ = 7.06, *p* = 0.854). Post hoc power for this analysis was 0.28, and the distribution of severity grades was similar across all four nutritional status groups. Detailed distribution of anaphylaxis severity by age and sex is provided in Appendix A. Sensitivity analyses with aggregated BMI categories (normal weight vs. overweight/obese), with or without collapsing severity into mild (grades 1–3) vs. severe (grades 4–5), yielded consistent, non-significant results (Appendix A). To further examine whether BMI was independently associated with anaphylaxis severity, we performed multivariate analyses. Both ordinal logistic regression (WAO grades 1–5 as outcome) and binary logistic regression (severe [grades 4–5] vs. non-severe [grades 1–3]) were adjusted for age, sex, asthma, and atopic dermatitis. In all models, BMI—whether expressed as a percentile or categorized into nutritional status groups—was not associated with anaphylaxis severity (all adjusted ORs close to 1, 95% CIs crossing 1, *p* > 0.05). Atopic dermatitis showed a protective association in the ordinal models (adjusted OR ≈ 0.55, *p* < 0.05), but not in the binary models, while age, sex, and asthma were not significant predictors in any model. Model fit was low (pseudo-R^2^ ≈ 0.02–0.03). Detailed outputs are provided in Appendix A.

### 3.3. Association Between Nutritional Status and Clinical Presentation by Organ System

Further analysis did not demonstrate a statistically significant difference in the frequency of symptoms for any of the five analyzed organ systems (cutaneous, respiratory, gastrointestinal, cardiovascular, and neurological) between the BMI categories (*p* > 0.05 for all tests). Although the overall results were not significant, borderline *p*-values were noted for respiratory, gastrointestinal, and cutaneous symptoms, which may suggest a potential trend that could become significant in a larger sample. The post hoc pairwise analysis identified only one comparison that met the corrected significance threshold (*p* < 0.0083): cutaneous symptoms were significantly less frequent in the underweight group compared to the normal weight group (*p* = 0.0057). However, as the omnibus test for cutaneous symptoms across all four groups was not significant, this isolated finding should be interpreted with caution (Figure 1).

### 3.4. Symptom Frequency Distribution Within BMI Categories

While no differences were found between the BMI groups, a separate analysis of the symptom distribution within each category was performed. For patients with normal weight, overweight, and obesity, this analysis showed that the frequency of symptoms from different organ systems was significantly different (Cochran’s Q test, *p* < 0.05). In these three groups, a consistent hierarchy was observed: cutaneous symptoms were the most frequent; respiratory and gastrointestinal symptoms occurred with moderate frequency; and cardiovascular and neurological symptoms were the least common. In contrast, for the underweight group, the same test did not reach statistical significance (*p* > 0.05). This indicates that a similar hierarchy of symptoms, while potentially present, could not be statistically confirmed, likely due to the small size of this subgroup (Table 2).

## 4. Discussion

The present study demonstrated no statistically significant association between nutritional status, as assessed by Body Mass Index (BMI) categories, and the clinical severity of anaphylaxis or the pattern of organ system involvement in our pediatric cohort. Baseline characteristics were homogenous across the compared groups with respect to key demographic and clinical variables, including age, sex, and the prevalence of comorbid atopic diseases, thus strengthening the validity of the primary finding by minimizing the influence of potential confounders. In multivariate analyses adjusted for age, sex, asthma, and atopic dermatitis, BMI was not identified as an independent predictor of anaphylaxis severity. This finding corroborates the results of the univariate tests and further supports the conclusion that nutritional status does not play a major role in shaping the clinical course of pediatric anaphylaxis. A protective association with atopic dermatitis was observed in some models, although this signal was inconsistent across analytic approaches and should be interpreted with caution. The low explanatory power of the regression models (pseudo-R^2^ ≈ 0.02–0.03) additionally suggests that factors beyond BMI and common atopic comorbidities—such as elicitor type, treatment timing, or co-exposures—are likely to be more relevant determinants of severity in this age group. The study did not find a statistically significant association between BMI categories and the frequency of respiratory symptoms, thus failing to confirm the initial hypothesis. However, a trend towards a higher frequency in the groups with higher BMI was observed, with a borderline statistical significance. This trend is biologically plausible and consistent with extant literature indicating the existence of mechanisms that may predispose children with obesity to more severe respiratory symptoms. Obesity has been demonstrated to exert a detrimental effect on respiratory mechanics by diminishing lung volumes, encompassing FRC and ERV [9]. Furthermore, the phenomenon of dysanapsis—a mismatch between airway diameter and lung volume—has been described in obese children and may lead to increased bronchial hyperreactivity [8]. In addition, obesity-related immune dysregulation, characterized by elevated pro-inflammatory cytokines and adipokine-driven modulation of immune cell function, provides further rationale for considering obesity a potential modifier of respiratory manifestations during anaphylaxis. It is therefore plausible that while a pathophysiological basis for more frequent respiratory symptoms exists, the trend observed in this study did not reach the threshold of statistical significance due to limited statistical power in this particular analysis.

The absence of a significant association between BMI and either the overall severity of anaphylaxis or the frequency of specific organ involvement in our pediatric cohort is at odds with data from the extant literature, which, based predominantly on adult populations, identifies obesity as a significant risk factor for more severe allergic reactions. A case–control study by Sadleir et al. demonstrated that obesity is a strong, independent risk factor for both the occurrence (OR = 2.96) and severity (OR = 2.61) of perioperative anaphylaxis in adults. The authors noted that this risk increased with higher grades of obesity (BMI grade) [10]. A similar conclusion was reached in a review on fatal anaphylaxis by Pouessel et al., which identified obesity as one of the most frequently recurring risk factors for mortality, while also noting that fatal outcomes are more common in the adult population [11]. The prevailing comprehension of the correlation between obesity and anaphylaxis severity is thus predominantly informed by research conducted in adult patient populations.

One potential explanation for this discrepancy is that the influence of obesity on anaphylaxis severity may be an age-dependent phenomenon. It is important to acknowledge that the pathophysiological basis for such an association is already evident in the pediatric population. The existing literature indicates that obesity in children is associated with a systemic low-grade inflammatory state, reflected in increased pro-inflammatory mediators and altered adipokine signaling, together with reduced anti-inflammatory activity [2,8]. This pro-inflammatory milieu is known to have direct clinical relevance in the course of asthma—a chronic allergic disease—where obesity is associated with more severe symptoms, poorer disease control, and a diminished response to treatment [8]. In adults, the long-term persistence of obesity-associated inflammation has been demonstrated to promote the development of cardiovascular comorbidities, which themselves are independent risk factors for adverse anaphylaxis outcomes [11]. Large cohort data indicate that pre-existing cardiac or respiratory disease is associated with higher odds of severe or near-fatal anaphylaxis, manifesting as increased rates of hospital or ICU admission and endotracheal intubation [14]. This underscores the prognostic weight of comorbidity burden in adult populations. Mechanistic evidence further supports this association: the cardiovascular system is a major determinant of anaphylaxis severity; cardiac mast cells—more numerous in ischemic heart disease and cardiomyopathies—release mediators (histamine, cysteinyl leukotrienes, prostaglandin D2, platelet-activating factor) that depress myocardial contractility, precipitate arrhythmias, and reduce coronary perfusion, lowering the threshold for hemodynamic collapse [15]. Furthermore, pharmacotherapies frequently utilized in these patients (β-blockers and angiotensin-converting enzyme inhibitors) have been associated with heightened anaphylaxis severity at the population level (pooled odds ratios ≈ 2.19 and 1.56, respectively), with the authors acknowledging the potential for residual confounding by underlying cardiovascular disease [16]. When considered as a whole, these data provide a coherent explanation for why obesity may act as a stronger modifier of anaphylaxis severity in adults, whereas in the pediatric population this compounding effect of established comorbidities is largely absent.

Furthermore, the phenotype of the most severe anaphylaxis appears to be less prevalent in this age group. As demonstrated in the systematic review by Turner et al. and the work by Pouessel et al., fatal anaphylaxis is reported significantly more often in adults and adolescents than in young children [17,18]. This age gradient likely reflects differences in elicitors and terminal pathways: food-triggered reactions predominate in childhood and more often culminate in respiratory failure, whereas drug- and venom-induced reactions are relatively more prevalent in older age groups, in whom cardiovascular terminal pathways are over-represented [17,18]. Accordingly, the relative scarcity of this phenotype in younger children provides a coherent explanation for the absence of a measurable BMI–severity signal in our pediatric cohort, despite biologically plausible links between excess adiposity and respiratory morbidity. Consequently, the following hypothesis can be formulated:

The consequences of childhood overweight and obesity, in the absence of established organ complications and combined with a lower baseline frequency of the most severe outcomes, may be insufficient to significantly modify the clinical severity of an acute anaphylactic reaction.

The analysis further revealed no significant differences in the severity of anaphylaxis among the underweight patient group. This finding is of particular interest in the context of recent evidence regarding the impact of malnutrition on the immune system. Current data suggest that malnutrition and micronutrient deficiencies do not lead to generalized immunosuppression, but rather to an immunological imbalance that promotes a pro-allergic, Th2-dominant response [19]. Although such an immune profile may create a predisposition for allergic sensitization, the findings of this study indicate that this does not necessarily translate into increased clinical severity during the effector phase of anaphylaxis in children.

The present study’s merits lie in its specific focus on a pediatric population, thereby addressing a critical knowledge gap in a field dominated by analyses of adults. Moreover, the incorporation of the entire range of nutritional status, from underweight to obese, facilitated a more comprehensive evaluation than studies that exclusively compare dichotomous groups. However, it is important to acknowledge the limitations of this work. The retrospective design precludes the establishment of causality and is susceptible to the inherent risk of incomplete data in medical records. The uneven distribution of sample sizes across the subgroups is also worthy of consideration; the reduced number of patients in the underweight and obese categories may have constrained the statistical power to detect more subtle associations. Furthermore, no a priori sample size calculation was performed, as all eligible cases from the study period were included. Post hoc power analyses were therefore carried out for the main and sensitivity analyses. None of the analyses demonstrated statistically significant associations, suggesting that while small or moderate effects in the smallest subgroups cannot be excluded, the study provides no evidence of a strong association between BMI and anaphylaxis severity in children. Finally, while the single-center design guarantees diagnostic and therapeutic consistency, it simultaneously restricts the generalizability of the findings to other patient populations.

## 5. Conclusions

Within the pediatric cohort under investigation, and in contrast to reports from adult populations, nutritional status, as measured by BMI, did not emerge as a significant predictor of anaphylaxis severity or the pattern of organ system involvement. The primary implication of this study is that the role of obesity as a modifying factor in the course of anaphylaxis may be age-dependent, and that findings from adult studies should be extrapolated to the pediatric population with caution. Further prospective, multicenter studies are warranted in this age group to confirm these observations and to better elucidate the complex interactions between nutritional status, age, and the underlying mechanisms of anaphylaxis.

## Figures and Tables

**Figure 1 nutrients-17-03023-f001:**
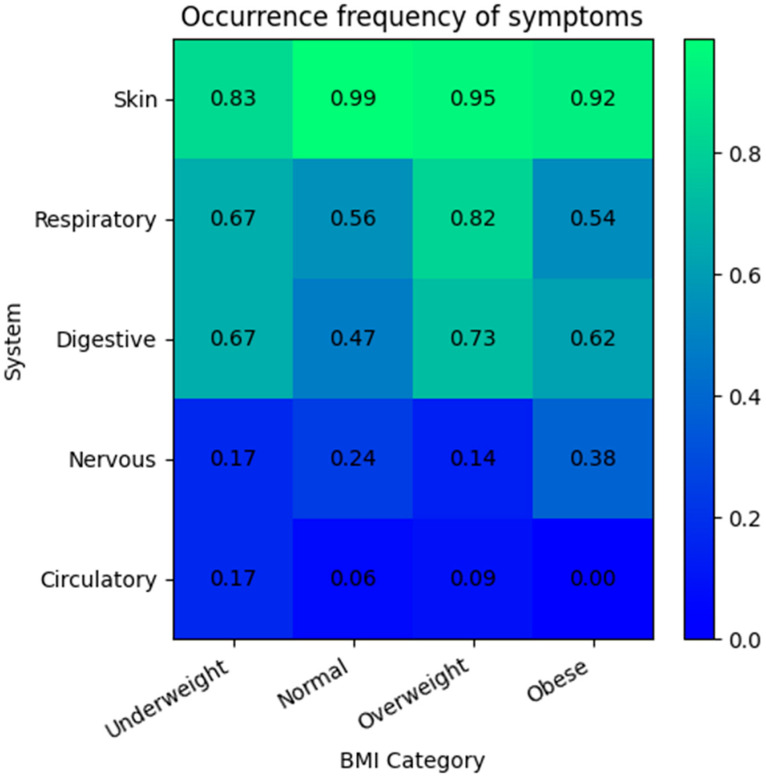
Frequency of clinical symptoms by organ system across BMI categories. The heatmap illustrates the proportion of patients in each BMI category who presented with symptoms from the respective organ systems. The color intensity corresponds to the frequency, with darker shades indicating a lower proportion of affected patients.

**Table 1 nutrients-17-03023-t001:** Demographic and clinical characteristics of the study cohort stratified by nutritional status.

Characteristic	Underweight (n = 6)	Normal Weight (n = 158)	Overweight (n = 22)	Obese (n = 13)	Total (N = 199)	*p*-Value
Age (years), mean ± SD ^1^	2.4 ± 2.9	4.2 ± 4.5	4.6 ± 4.9	4.5 ± 5.9	4.2 ± 4.6	0.519
Sex, n (%)						0.096
Male	5 (83.3)	104 (65.8)	18 (81.8)	12 (92.3)	139 (69.8)	
Female	1 (16.7)	54 (34.2)	4 (18.2)	1 (7.7)	60 (30.2)	
Asthma, n (%)	0 (0.0)	34 (21.5)	6 (27.3)	5 (38.5)	45 (22.6)	0.261
Atopic Dermatitis, n (%)	2 (33.3)	75 (47.5)	10 (45.5)	8 (61.5)	95 (47.7)	0.673

^1^ SD, Standard Deviation.

**Table 2 nutrients-17-03023-t002:** Distribution of clinical manifestations of anaphylaxis across BMI categories. Values are presented as number of patients (% within each BMI group). Multiple organ systems could be involved in a single patient.

BMI Category	Cardiovascular	Respiratory	Skin	Digestive	Neurological
Underweight (n = 6)	1 (16.7)	4 (66.7)	5 (83.3)	4 (66.7)	1 (16.7)
Normal weight (n = 158)	10 (6.3)	88 (55.7)	156 (98.7)	75 (47.5)	38 (24.1)
Overweight (n = 22)	2 (9.1)	18 (81.8)	21 (95.5)	16 (72.7)	3 (13.6)
Obese (n = 13)	0 (0.0)	7 (53.8)	12 (92.3)	8 (61.5)	5 (38.5)

## Data Availability

The minimal dataset generated and analyzed during the current study is available in the Appendix A.

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
