# Peer review of "Nutritional Status Is Not a Predictor of Anaphylaxis Severity in a Pediatric Cohort: A Retrospective Analysis"

_nutrients, 2025, doi:10.3390/nu17183023_

Round 1

Reviewer 1 Report

Comments and Suggestions for Authors

Evaluation of Manuscript nutrients-3873532

This is an interesting research topic, as there are few studies on the subject, and the authors have conducted it with scientific rigor. I will present my considerations on the topic below.

The study is robust and has a relevant and adequate sample size.

The title is appropriate and the Abstract is well-constructed; however, the results are poorly presented. I believe the authors could better explore the Table and the Figure in the Abstract.

The study's introduction lacks a clear narrative line, jumping from the prevalence of anaphylaxis to food allergens, obesity, its inflammatory mechanisms, and asthma, before finally presenting the central research question about the relationship between obesity and anaphylaxis. This long sequence with unnecessary topics confuses the reader instead of properly presenting the study's idea and relevance. The inclusion of irrelevant information exacerbates the problem: the mention of food allergens is superfluous, as the study does not investigate the triggers of the reaction, and the detailed discussion of inflammatory biomarkers like IL-6 and leptin is speculative, since the methodology does not include the measurement of such markers, making it a point better suited for the discussion section. As a result, the research question is only presented at the end of the second-to-last paragraph. This means the reader is faced with extensive generic context before understanding the true objective of the work. In other words, the introduction does not truly present what the authors want to investigate, why it is being investigated, and for what purpose. Please rewrite it, and at the end of the introduction, present the study's hypothesis.

Regarding the methods, a notable methodological limitation is the absence of a sample size calculation. The study used a convenience sample based on the number of available cases. I believe there should be no issue in performing this calculation, as the number of included cases is large. However, without a power calculation, it is impossible to determine if the study had an adequate sample to detect statistically significant differences, should they truly exist. This calculation is necessary not only for the overall sample size but also for the subgroups; there are 158 normal-weight, 22 overweight, and 13 obese individuals. The study's main question is precisely the relationship between nutritional status and severe anaphylaxis. In this context, it is very important for the authors to demonstrate that the 13 (6.5%) obese and 6 (3%) underweight participants are representative enough to answer the research questions. Otherwise, the obese individuals will have to be combined with the overweight group, and the underweight individuals could be combined with the normal-weight group or excluded to increase statistical power.

It is also necessary to describe the purpose of certain measurements. The results mention asthma and dermatitis, which are not part of the EAACI 2021 diagnostic criteria. In this regard, the authors need to be more detailed about the diagnosis; as it is not a very common topic, readers may find it difficult to understand the manuscript. As the methods are currently written, it is not possible to reproduce the study or even understand how the authors arrived at their results. Finally, the exclusion criteria are also not clear.

Regarding the statistics, I have another suggestion for the authors, as the chosen tests are not sufficient for the desired answers. Although the univariate analysis presented shows no association between BMI and anaphylaxis severity, the absence of a multivariate analysis prevents a robust conclusion. In Table 1, asthma and dermatitis are presented as confounding variables that were controlled for. However, to state with greater confidence that nutritional status is not an independent predictor of reaction severity, it is essential to perform a logistic regression analysis that simultaneously adjusts for variables such as age, sex, asthma, and atopic dermatitis. This approach would isolate the specific effect of BMI, and the persistence of a non-significant odds ratio after this adjustment would greatly strengthen the validity of the main result. Therefore, I suggest the authors implement a model that dichotomizes the outcome into "severe anaphylaxis" (WAO 4-5) versus "non-severe" (WAO 1-3) or that uses the full ordinal scale of the WAO. The results, presented as adjusted odds ratios with confidence intervals, would allow for a clearer and more causally informative interpretation of the relationship under investigation.

The results are adequate for the current statistical model but will need to be redone based on my suggestion above.

The discussion should also be reworked. I recommend that the authors avoid in-depth discussion of variables that were not measured in the present study. There is a long paragraph discussing pro-inflammatory cytokines. I recommend that this be revised.

Reviewer 2 Report

Comments and Suggestions for Authors

Dear authors,

I have read with interest your manuscript, and I send you my comments:

1) Please, in Table 1, are you sure that the difference between male and female is not significant? i.e., in obese patients, are you sure that there is not a significant difference between the number of male vs. female?

2) Section 3.2: Please add the data of anaphylaxis reported for each patient enrolled with the difference for age and for gender.

3) Please describe in a table the clinical symptoms represented in section 3.4 and figure 1 for each group's characteristics: underweight, normal weight, overweight, and obese.

Round 2

Reviewer 1 Report

Comments and Suggestions for Authors

I realize I was quite demanding with the authors in the first draft of the manuscript, and I received an appropriate response. I believe the text is now ready for publication; however, authors must fully format the text to the journal's standards.

Reviewer 2 Report

Comments and Suggestions for Authors

none